# Experimental Hut Trials Reveal That CYP6P9a/b P450 Alleles Are Reducing the Efficacy of Pyrethroid-Only Olyset Net against the Malaria Vector *Anopheles funestus* but PBO-Based Olyset Plus Net Remains Effective

**DOI:** 10.3390/pathogens11060638

**Published:** 2022-06-01

**Authors:** Benjamin D. Menze, Leon M. J. Mugenzi, Magellan Tchouakui, Murielle J. Wondji, Micareme Tchoupo, Charles S. Wondji

**Affiliations:** 1Vector Biology Department, Liverpool School of Tropical Medicine, Pembroke Place, Liverpool L3 5QA, UK; murielle.wondji@lstmed.ac.uk; 2Medical Entomology Department, Centre for Research in Infectious Diseases (CRID), Yaoundé 13591, Cameroon; leon.mugenzi@crid-cam.net (L.M.J.M.); magellan.tchouakui@crid-cam.net (M.T.); micareme.tchoupo@crid-cam.net (M.T.)

**Keywords:** malaria, long-lasting insecticidal nets, insecticide resistance, metabolic resistance, cytochrome P450, *Anopheles funestus*

## Abstract

Malaria remains a major public health concern in Africa. Metabolic resistance in major malaria vectors such as *An. funestus* is jeopardizing the effectiveness of long-lasting insecticidal nets (LLINs) to control malaria. Here, we used experimental hut trials (EHTs) to investigate the impact of cytochrome P450-based resistance on the efficacy of PBO-based net (Olyset Plus) compared to a permethrin-only net (Olyset), revealing a greater loss of efficacy for the latter. EHT performed with progenies of F5 crossing between the *An. funestus* pyrethroid-resistant strain FUMOZ and the pyrethroid-susceptible strain FANG revealed that PBO-based nets (Olyset Plus) induced a significantly higher mortality rate (99.1%) than pyrethroid-only nets (Olyset) (56.7%) (*p* < 0.0001). The blood-feeding rate was higher in Olyset compared to Olyset Plus (11.6% vs. 5.6%; *p* = 0.013). Genotyping the *CYP6P9a/b* and the intergenic *6.5 kb structural variant* (SV) resistance alleles showed that, for both nets, homozygote-resistant mosquitoes have a greater ability to blood-feed than the susceptible mosquitoes. Homozygote-resistant genotypes significantly survived more with Olyset after cone assays (e.g., *CYP6P9a* OR = 34.6; *p* < 0.0001) than homozygote-susceptible mosquitoes. A similar but lower correlation was seen with Olyset Plus (OR = 6.4; *p* < 0.001). Genotyping EHT samples confirmed that *CYP6P9a/b* and *6.5 kb_SV* homozygote-resistant mosquitoes survive and blood-feed significantly better than homozygote-susceptible mosquitoes when exposed to Olyset. Our findings highlight the negative impact of P450-based resistance on pyrethroid-only nets, further supporting that PBO nets, such as Olyset Plus, are a better solution in areas of P450-mediated resistance to pyrethroids.

## 1. Introduction

Long-lasting insecticidal nets (LLINs) remain an important component of the malaria control strategies used to reduce mortality and morbidity due to this disease in Africa [1,2]. Mostly insecticides belonging to the pyrethroid group, as well as more and more novel insecticides, are currently recommended for bed net impregnation [3,4,5]. In recent years, anophelines mosquitoes have increasingly been reported to show resistance against pyrethroids across Africa. This is the case for major vectors including *Anopheles funestus* [6,7,8,9] and *Anopheles gambiae* [10,11,12,13,14]. This growing spread of resistance has led to a concern that LLINs may lose their efficacy. However, quantifying such loss of efficacy remains challenging without suitable metrics associated with resistance, highlighting the need to use robust genotype/phenotype analysis for this evaluation. At a time when national malaria control programs across Africa are seeking to take evidence-based decisions on their choice of suitable nets to help improve malaria control, it is paramount to determine the extent to which existing resistance to pyrethroid really affects LLIN efficacy. This involves an understanding of the interaction between resistance mechanisms and mosquito responses to exposure to LLINs. Broadly speaking, pyrethroid resistance can be caused by alterations in the target site of the insecticide or enhanced enzymatic activities capable of metabolizing the insecticide before it reaches the target [15,16,17]. Target site resistance is well studied with mutations in the target of both pyrethroid and DDT insecticides, as well as the voltage-gated sodium channels which lead to knockdown resistance (*kdr*). The development of molecular assays more than twenty years ago made it possible to track this resistance mechanism using a simple PCR [18,19,20,21] or high throughput techniques, such as TaqMan assays [22]. It has also been possible to assess the impact of *kdr* on control tools, notably in *An. gambiae* where these markers are common [23,24,25], contrary to *An. funestus*, where it is still not detected [26]. The second best characterized cause of insecticide resistance is metabolic resistance, which is more complex to understand as it can involve the detoxification, sequestration, and transportation of insecticides or their conjugates [16,17,27]. Recent studies have focused on elucidating the genetic factors which cause the increased expression of detoxification genes associated with insecticide resistance in malaria vectors such as *An. funestus*, leading to the development of simple molecular assays used to detect metabolic resistance [28,29,30,31].

The recent design of molecular assays for the duplicated *CYP6P9a/b* and the associated *6.5 kb* structural variant (SV), refs. [28,29,31] now provides a great opportunity to assess the impact of P450-based pyrethroid resistance on the efficacy of various LLINs, including pyrethroid-only (Olyset) and PBO (Olyset Plus) nets. This will provide evidence-based information on their effectiveness in the presence of a growing and escalading resistance to pyrethroids. Previous assessment of this impact on PermaNet nets made with type II pyrethroid deltamethrin has been performed [28,29,31], but this remains to be carried out for type I pyrethroid nets, such as the common permethrin-based Olyset nets. 

Here, using EHT, we show a significantly greater efficacy of the PBO-based net Olyset Plus compared to the pyrethroid-only Olyset net against pyrethroid-resistant *An. funestus.* Using the genotyping of DNA-based markers of P450 resistance, we reveal that *CYP6P9a/b* P450-based pyrethroid resistance induces a significant loss of efficacy for pyrethroid-only net Olyset against *An. funestus*. A reduced efficacy was also detected for PBO-based net Olyset Plus, but at a lower extent than for Olyset net, revealing that PBO nets are more suitable to tackle such P450-based resistance.

## 2. Results

### 2.1. Insecticide Susceptibility of Lab Strain and Crosses with WHO Cone and Tube Test

#### 2.1.1. Quality Control and Performance of the Nets against the Hybrid Strain

All the nets used in the study were first exposed to the susceptible lab strain Kisumu for quality control. The mortality for Olyset and Olyset Plus was 100% (Figure 1A). The *An. funestus* hybrid strain exposed to bed nets using WHO cone assays exhibited mortality rates of 30.1 ± 11.5% for Olyset and 100 ± 0% for Olyset Plus, suggesting a greater efficacy of PBO synergist nets compared to pyrethroid-only nets.

**Susceptibility profiles of the FUMOZ-FANG:** The bioassays performed with the reciprocal FUMOZ-FANG strains revealed that the hybrid strain was resistant to deltamethrin with mortality of 43.7 ± 5.9%, and was resistant to propoxur with mortality of 67.3 ± 6.6% (Figure 1B). The second set of bioassays performed with the reciprocal FUMOZ-FANG strains revealed that the hybrid strain was resistant to permethrin with mortality of 43.7 ± 1.6% and 39.3 ± 1.6%, respectively, for 90 min and 30 min. Resistance was also observed with deltamethrin with a mortality of 86.4 ± 3.1% and 42.3 ± 2.5%, respectively, for 90 min and 30 min (Figure 1C).

#### 2.1.2. Performance of Nets against the Hybrid Strain Using Experimental Hut

A total number of 578 mosquitoes from the hybrid strain were released and recaptured on a period of one week. In total, 141 samples were released in the control hut, 224 in the hut were treated with Olyset, and 213 in the hut were treated with Olyset Plus. 

**Mortality**: Analysis of mortality rates revealed very high mortality of the hybrid FUMOZ-FANG strain against the PBO-based Olyset Plus net (99.1%). In contrast, lower mortality was observed for the pyrethroid-only Olyset net (56.7%). Very low mortality was observed in the untreated control net (9.9%) (Figure 1D; Appendix A). 

**Blood-feeding**: The blood-feeding rate did not significantly differ when comparing the control (7.8%) to Olyset (11.6% *p* ˃ 0.05) and Olyset Plus (5.6%; *p* ˃ 0.05). However, the blood-feeding rate was higher in Olyset compared to Olyset Plus (11.6% vs. 5.6%; *p* = 0.013) (Figure 1D; Appendix A).

**Exophily**: The exophily rate in the hut with Olyset net (23.7%) was significantly higher than that in the control hut (13.5%) (*p* = 0.02), as well as than for Olyset Plus (7.5%; *p* < 0.001). No significant difference was observed between Olyset Plus and the control net (*p* = 0.3) (Figure 1D; Appendix A). 

### 2.2. Validating the Role of CYP6P9a/b and 6.5 kb SV in Pyrethroid Resistance in the Hybrid FUMOZ-FANG Strains before the Experimental Hut Trials

#### 2.2.1. Role of CYP6P9a_R Using WHO Tube Assay Samples

Using the hybrid FUMOZ-FANG strains, the role of the *CYP6P9a*_R allele in the observed pyrethroid resistance was confirmed. The odds ratio of surviving exposure to permethrin when homozygous for the resistant *CYP6P9a*_R allele (RR) was high at 693 (CI 88–5421; *p* < 0.0001) compared to the homozygous-susceptible mosquitoes (SS) (Figure 2A). The OR was 131 (CI 27–978; *p* < 0.0001) when comparing RR to RS, indicating the increasing resistance conferred by *CYP6P9a*.

#### 2.2.2. Validating the Role of CYP6P9b_R Using Sample from WHO Tube Assays

To confirm the ability of the *CYP6P9b*_R allele to predict pyrethroid resistance phenotype in the hybrid strain FUMOZ-FANG, the F5 samples were genotyped. This revealed that those that stayed alive after exposure to permethrin for 90 minutes are mainly homozygote-resistant (20.8%) and heterozygotes (75%), with only two being homozygote-susceptible. Among the 47 mosquitoes found dead after 30 min exposure to permethrin (highly susceptible), 97.87% were found to be homozygote-susceptible, and one remaining was a heterozygote (Figure 2B). A strong association was perceived between permethrin resistance and the *CYP6P9b* genotypes when comparing RR vs. SS (OR = infinity; *p* < 0.0001) and RS vs. SS (OR = 715; *p* < 0.0001).

#### 2.2.3. Validation of the Role of CYP6P9a and CYP6P9b in Conferring Resistance Using Samples from Cone Assays

Using dead and alive mosquitoes obtained after cone assays, *CYP6P9a* homozygous-resistant mosquitoes (RR) and heterozygous (RS) could survive significantly better following exposure to Olyset and Olyset Plus than homozygote-susceptible mosquitoes (Appendix A). Olyset shows a strong association between the resistance alleles and the ability to survive when comparing RR vs. SS (OR = 35.1; *p* < 0.0001) and RS vs. SS (OR = 34.6; *p* < 0.0001) for *CYP6P9a* (Figure 2C). The same trend was observed when looking at the allelic frequency R vs. S (OR = 7.7; *p* < 0.05) (Appendix A; Figure 2D). Concerning *CYP6P9b*, homozygous-resistant mosquitoes (RR) and heterozygous (RS) could also survive significantly better following survive exposure to Olyset net. A strong association was noticed between the gene and the ability to survive when comparing RR to SS (OR = 32.4; *p* < 0.0001) and RS vs. SS (OR = 97.3; *p* < 0.0001) (Appendix A; Figure 2E).

Olyset Plus shows a strong association between the resistance alleles and the ability to survive when comparing RR vs. SS (OR = 6.4; *p* < 0.001) and RS vs. SS (OR = 7.2; *p* < 0.001) for *CYP6P9a* (Appendix A; Figure 3A). The same trend was observed when looking at the allelic frequency R vs. S (OR = 2.0; *p* = 0.05) (Appendix A; Figure 3B). Concerning the *CYP6P9b*, Olyset Plus equally shows a strong association between the resistance alleles and the ability to survive when comparing RR vs. SS (OR = 9.7; *p* < 0.001) and RS vs. SS (OR = 17.9; *p* < 0.001) for *CYP6P9b* (Appendix A; Figure 3C). The same trend was observed when looking at the allelic frequency R vs. S (OR = 2.2; *p* = 0.01) (Appendix A).

#### 2.2.4. Validating the Role of 6.5 kb SV in Conferring Resistance Using Samples from Cone Assays

To validate the ability of the *6.5 kb SV* marker to predict the impact of resistance on the efficacy of LLINs, we genotyped F5 samples obtained from cone assays with Olyset and Olyset Plus. Comparing the proportion of each SV genotype between alive and dead mosquitoes revealed that SV+/SV+ homozygote mosquitoes were significantly more likely to survive exposure to Olyset than those completely lacking the SV (SV−/SV−) (OR = 52.6; CI = 17.7–161.7; *p* < 0.00013) (Appendix A; Figure 4A). Heterozygotes could also survive better following exposure to Olyset compared to homozygotes which lacked the SV (SV−/SV−) (OR = 27.5; CI = 11.42–66.3; *p* < 0.0001). SV+/SV+ homozygous mosquitoes could not survive as well as heterozygotes (OR = 1.91; CI = 0.6–5.3; *p* = 0.1). When drawing comparisons at the allelic level, it was observed that possessing a single SV+ allele provides the significant ability to survive against exposure to Olyset compared to the SV− allele (OR = 9.9; CI = 5.05–19.8; *p* < 0.0001) (Appendix A; Figure 4B). A similar trend was observed with Olyset Plus when drawing comparisons at the allelic level, it was observed that possessing a single SV+ allele provides the significant ability to survive against exposure to Olyset plus compared to the SV− allele (OR = 7.1; CI = 3.4–14.8; *p* < 0.0001) (Appendix A; Figure 4C).

### 2.3. Impact of CYP6P9a on the Efficacy of Olyset and Olyset plus Nets Using Experimental Hut Trial

#### 2.3.1. Impact on Mosquito Mortality

Genotyping of the *CYP6P9a* marker allowed us to assess the impact of P450-based metabolic resistance on the loss of efficacy of Olyset, but not for Olyset Plus, since most of the mosquitoes released in Olyset Plus huts died. To avoid confounding effects from blood-feeding, or net entry or exophily status, the distribution of the *CYP6P9a* genotypes was assessed firstly only among unfed mosquitoes collected in the room. This revealed a highly significant difference in the frequency of the three genotypes between the dead and alive mosquitoes (chi-square = 28.8; *p* < 0.0001) (Figure 5). Analysis of the correlation between each genotype and mortality revealed that *CYP6P9a* homozygous-resistant mosquitoes (RR) show an increased ability to survive exposure to the Olyset when compared to homozygote-susceptible mosquitoes (SS) with (OR = 5.0; CI = 2.01–12.4; *p* = 0.001). The strength of this association was further shown by the significant ability to survive exposure when possessing a single *CYP6P9a*-resistant allele (R) compared to the susceptible allele (S) (OR = 2.04; CI = 1.1–3.6; *p* < 0.05). However, heterozygote mosquitoes (RS) did not survive exposure to Olyset better than homozygote-susceptible mosquitoes (SS) (OR = 1.8; CI = 0.8–3.9; *p* ˃ 0.05) (Figure 5A; Appendix A; Figure 5B).

#### 2.3.2. CYP6P9a Impacting the Blood-Feeding

Homozygote *CYP6P9a*-resistant mosquitoes (RR) were significantly more likely to blood-feed better than homozygote-susceptible mosquitoes (SS) (OR = 4.5; *p* < 0.01) when exposed to Olyset. Heterozygote mosquitoes (RS) were significantly worse at blood-feeding with Olyset compared to homozygote-susceptible mosquitoes (SS) (OR = 0.8; CI = 0.3–1.7; *p* = 1). Homozygote-resistant mosquitoes (RR) fed heterozygotes (OR = 5.4; CI = 2.6–10.88; *p* < 0.001) significantly better (Appendix A; Figure 5C). Overall, the strength of this association was further shown by the significant ability to blood-feed when possessing a single *CYP6P9a*-resistant allele (R) compared to the susceptible allele (S) (OR = 2.3; CI = 1.3–4.1; *p* < 0.01) (Appendix A). For Olyset Plus, homozygote *CYP6P9a*-resistant mosquitoes (RR) were significantly more likely to blood-feed than homozygote-susceptible mosquitoes (RS) (OR = 6.3; *p* < 0.001) when exposed to Olyset Plus. The significant ability to blood-feed was observed when possessing a single *CYP6P9a*-resistant allele (R) compared to the susceptible allele (S) (OR = 2.0; CI = 1.1–3.5; *p* = 0.03) (Appendix A). 

### 2.4. Impact of CYP6P9b on the Efficacy of Olyset and Olyset plus Nets Using Experimental Hut Trial

#### 2.4.1. The Impact of CYP6P9b on Mortality

The impact of *CYP6P9b* genotypes on the ability of mosquitoes to survive exposure to Olyset was firstly evaluated on the unfed samples collected only in the room. This investigation showed that the three genotypes were significantly different in their distributions (chi-square = 23.4; *p* < 0.0001) (Figure 5). Analysis of the correlation between each genotype and mortality revealed that *CYP6P9b* homozygous-resistant mosquitoes (RR) show an increased ability to survive exposure to the Olyset when compared to homozygote-susceptible mosquitoes (SS) (OR = 15.0; CI = 4.5–50.3; *p* < 0.001) (Figure 5D). Heterozygote mosquitoes (RS) survived exposure to Olyset significantly better than homozygote-susceptible mosquitoes (SS) (OR = 6; CI = 1.9–18.75; *p* < 0.001). Homozygote-resistant mosquitoes (RR) struggled to survive compared to heterozygotes (OR = 2.4; CI = 1.2–4.8; *p* = 1) when considering samples from the room only. On the other hand, when considering all the samples, the additive resistance conferred by each allele of *CYP6P9b* was shown by the fact that homozygote-resistant mosquitoes (RR) could survive significantly better than heterozygotes (OR = 2.5; CI = 1.3–4.8; *p* < 0.01). Moreover, the strength of this association was further shown by the significant ability to survive exposure when possessing a single *CYP6P9b*-resistant allele (R) compared to the susceptible allele (S) (OR = 2.5; CI = 1.4–4.5; *p* < 0.01) (Appendix A; Figure 5E). The same trend showing a strong association between the *CYP6P9b* genotypes and the mortality was also observed when analyzing all the samples dead and alive, including the blood-fed mosquitoes and the one collected in the veranda (Appendix A).

#### 2.4.2. CYP6P9b Impacting the Blood-Feeding

A strong association was observed between *CYP6P9b* genotypes and the ability to blood-feed. Homozygote-resistant mosquitoes (RR) could more successfully blood-fed when compared to homozygote-susceptible mosquitoes (SS) (OR = 10.01; CI = 4.1–24.9; *p* < 0.001) (Appendix A; Figure 5F). The blood-feeding ability of heterozygote mosquitoes (RS) in the presence of Olyset was not different to that of homozygote-susceptible mosquitoes (SS) (OR = 0.8; CI = 0.3–2.2; *p* = 1). Homozygote-resistant mosquitoes (RR) could blood-feed significantly better than heterozygotes (OR = 11.1; CI = 5.3–23.03; *p* < 0.001), suggesting an additive effect. Overall, the strength of this association was further shown by the significant ability to blood fed when possessing a single *CYP6P9b*-resistant allele (R) compared to the susceptible allele (S) (OR = 4.7; CI = 2.5–8.1; *p* < 0.01) (Appendix A).

### 2.5. Impact of 6.5 kb SV on the Efficacy of Olyset and Olyset plus Nets Using Experimental Hut Trial

#### 2.5.1. The Impact of 6.5. kb SV on Mortality

The impact of *6.5 kb SV* genotypes on the ability of mosquitoes to survive exposure to Olyset was firstly evaluated on the unfed samples collected only in the room. This showed that the three genotypes were significantly different in their distributions (*p* < 0.0001) (Figure 6). Analysis of the correlation between each genotype and mortality revealed that *6.5 kb* SV homozygous-resistant mosquitoes (RR) show an increased ability of the mosquitoes to survive exposure to the Olyset when compared to homozygote-susceptible mosquitoes (SS) (OR = 6.94; CI = 2.8–16.7; *p* < 0.0001) (Appendix A). Heterozygote mosquitoes (RS) survived exposure to Olyset better than homozygote-susceptible mosquitoes (SS) but not significantly (OR = 1.6; CI = 0.7–3.6; *p* = 0.2). The additive resistance conferred by each allele of *6.5 kb SV* was shown by the fact that homozygote-resistant mosquitoes (RR) could significantly survive more than heterozygotes homozygote-resistant mosquitoes (RR) (OR = 4.2; CI = 2.1–8.5; *p* < 0.0001) (Figure 6A) when considering sample from the room only and all the samples (Appendix A). Moreover, the strength of this association was further shown by the significant ability to survive exposure when possessing a single *6.5 kb SV*-resistant allele (R) compared to the susceptible allele (S) (OR = 2.7; CI = 1.5–4.9; *p* = 0.001) (Appendix A; Figure 6B). The same trend showing a strong association between the *6.5 kb SV* genotypes and the mortality was also observed when analyzing all the samples dead and alive including the blood-fed and the one collected in the veranda (Appendix A).

#### 2.5.2. The Impact of 6.5. kb SV on Blood-Feeding 

A strong association was observed between *6.5 kb* SV genotypes and the ability to blood-feed. Homozygote-resistant mosquitoes (RR) could blood-feed better compared to homozygote-susceptible mosquitoes (SS) (OR = 10.01; CI = 3.2–31.4; *p* < 0.0001) (Appendix A; Figure 6C). The blood-feeding ability of heterozygote mosquitoes (RS) in the presence of Olyset differs to that of homozygote-susceptible mosquitoes (SS) but not significantly (OR = 1.7; CI = 0.5–5.4; *p* = 0.4) (Appendix A). Homozygote-resistant mosquitoes (RR) could blood-feed significantly better than heterozygotes (OR = 10.2; CI = 4.9–21.11; *p* < 0.0001), suggesting an additive effect. Overall, the strength of this association was further shown by the significant ability to blood-feed when possessing a single *CYP6P9b*-resistant allele (R) compared to the susceptible allele (S) (OR = 4.2; CI = 2.3–7.9; *p* < 0.0001) (Appendix A; Figure 6D). A similar trend was observed against Olyset Plus nets (Appendix A; Figure 6E,F).

### 2.6. Combined Impact of CYP6P9a and CYP6Pb on the Efficacy of Olyset and Olyset plus Nets Using Experimental Hut Trial

As *CYP6P9b* genotypes were shown to be independent from those of *CYP6P9a* [28], we also assessed how combinations of genotypes at both genes impact the efficacy of Olyset for mortality and blood-feeding. Analysis of the impact of combined genotypes on mortality with Olyset confirmed the independent segregation of genotypes at both genes with several combinations of genotypes observed including RR/RR, RR/RS, RS/RS, RS/SS and SS/SS (Appendix A). A comparison of the distribution of both sets of genotypes revealed that double homozygote-resistant (RR/RR) mosquitoes at both genes had a far greater ability to survive exposure to Olyset than most of the other combinations (Appendix A) particularly when compared to double susceptible (SS/SS) (OR = 8.6; CI = 1.8–39.1; *p* < 0.01) (Appendix A). A significantly increased survival is also observed in RR/RR when compared to other combinations although with a lower odds ratio, such as against RR/SS (OR = 2; *p* < 0.01) (Appendix A).

Analysis of the combined genotype distribution for blood-feeding also revealed a significantly increased ability to blood-feed for double homozygote-resistant mosquitoes when exposed to Olyset with the highest significance against double homozygote-susceptible mosquitoes (OR = 9.3; *p* < 0.001) (Appendix A).

### 2.7. Combined Impact of the Triple Resistance Alleles CYP6P9a/CYP6P9b/6.5 kb SV on the Efficacy of Olyset and Olyset plus Nets Using Experimental Hut Trial

Experimental hut trials showed that triple RR/RR/SV^+^SV^+^ homozygotes were more likely to survive exposure to Olyset compared to any other combined genotypes except RR/RS/RS (Figure 7A). RR/RR/SV+SV+ survived more than SS/SS/SV−SV− (OR = 8.5; CI: 2.53–28.46; *p* = 0.0005) and RS/RS/SV+SV− (OR = 3.57; CI = 1.82– 7.02; *p* = 0.0002). There was no significant difference between the RR/RR/SV+SV+ and RR/RR/SV+SV− genotypes (OR = 1; CI = 0.16–7.02; *p* = 1.0), and both had a similar trend in terms of survival with the same odds ratio (Figure 7B). In addition to the ability to survive exposure to the net, RR/RR/SV+SV+ also could blood-feed better than SS/SS/SV− SV− (OR = 7.0; CI: 2.38–20.52; *p* = 0.0004) and the other combinations in the presence of Olyset (Figure 7C,D).

## 3. Discussion

The capacity to assess the impact of insecticide resistance on the effectiveness of insecticide-treated control tools is crucial in order to implement suitable insecticide resistance management (IRM) [32]. Evolution in the area of DNA-based resistance markers of P450-mediated resistance [28,29] currently offers robust tools to screen pyrethroid resistance in field populations of the major malaria vectors such as *An. funestus* and assess their impact on the effectiveness of LLINs. In this study, we use these markers to evaluate the extent to which pyrethroid resistance is impacting the efficacy of pyrethroid-only nets such as Olyset in comparison to PBO-based net such as Olyset Plus. This study also allowed to assess the interplay between P450 genes in the overall genetic variance to resistance and their combined impact on the efficacy of LLINs.

### 3.1. PBO-Based Nets (Olyset Plus) Exhibit Greater Efficacy than Pyrethroid-Only Nets (Olyset) in the Context of P450-Based Resistance

Olyset presented a moderate mortality compared to Olyset Plus (30.9% vs. 100%; *p* < 0.0001). The high mortality observed with PBO nets compared to pyrethroid-only nets is similar to what was observed in previous studies [33,34,35,36]. The high mortality observed with PBO-based nets against the hybrid strain FUMOZ/FANG is due to the fact the mechanisms underlying the resistance in this population are driven by *CYP6P9a* and *CYP6P9b* [37] which are inhibited by PBO [38]. These results demonstrate that PBO-based nets may be a more suitable solution in areas where resistance is mainly mediated by P450 genes. The efficacy of nets observed in this study confirms the loss in efficacy of the pyrethroid-only nets against *Anopheles* mosquitoes, as observed across the continent [39]. This loss of efficacy was observed in Mozambique [7,40], Malawi [41], Congo [42], and Cameroon [6,43]. Overall, similar to this study, it has been noticed that PBO-based nets demonstrate a better performance compared to pyrethroid-only nets [33,34]. The same trends were observed with pyrethroid type II with high mortality observed with PermaNet 3.0 compared to PermaNet 2.0 [28,29,31], suggesting that PBO-based nets with both type I or II can exhibit high performance against resistance sustained by P450s.

### 3.2. P450 Resistance Can Reduce the Efficacy of Pyrethroid-Only Nets Significantly Better Than PBO-Based Nets

When comparing the impact of *CYP6P9a/b* on pyrethroid-only nets and PBO-based nets using samples from cone assays, we noticed that the impact of *CYP6P9a* on pyrethroid-only nets was higher than with PBO-based nets (Olyset, OR = 35.1; *p* < 0.00001 vs. Olyset Plus, OR = 6.4; *p* < 0.001). The same trend was also observed with *CYP6P9b* and the 6.5 kb SV. This can be explained by the fact that the addition of the PBO inhibits the cytochrome P450 enzymes, which is the main resistance mechanism in the strain used by the mosquitoes [44,45].

On the other hand, Olyset nets have also shown reduced performance against *CYP6P9a*- and *CYP6P9b*-resistant mosquitoes. Strong association was observed between the resistant alleles and the increased ability of the mosquitoes to survive after exposure to these nets. Nevertheless, the impact of *CYP6P9a* and *CYP6P9b* seems to be more significant on PermaNet nets impregnated with deltamethrin [29,31], compared to Olyset nets impregnated with permethrin (Olyset, OR = 35.1; *p* < 0.00001 vs. PermaNet 2.0, OR = 239.0; *p* < 0.001 and Olyset Plus, OR = 6.4; *p* < 0.001 vs. PermaNet 3.0, OR = 81.0; *p* < 0.001) [29,31]. This significant impact on PermaNet could be associated with the fact that the hybrid strain is more resistant to type II pyrethroids, as shown by WHO bioassays, where the mosquitoes were more resistant to deltamethrin compared to permentrin (48.5% vs. 80.7%). With regards to the obtained odd ratios, a stronger association was observed between *CYP6P9a/CYP6P9b* and the loss of efficacy of nets compared to what was observed with GST-based resistance through the L119F-*GSTe2*-resistant allele [43], in line with the greater role of P450 in resistance to pyrethroids in *An. funestus*, compared to that of GST. The 6.5 kb SV was shown to further exacerbate the loss of efficacy in the permethrin-only nets. The design of the simple PCR-based assay to genotype the *6.5 kb SV* enabled us to assess the impact of such structural variation on the efficacy of insecticide-treated nets, including the pyrethroid-only and the PBO-synergist nets. A greater reduction in the efficacy of *6.5 kb SV* was present on permethrin-only nets compared to PBO-based nets, which is similar to that observed with *CYP6P9a* [31] and *CYP6P9b* [29], in terms of the reduced mortality rate and blood-feeding inhibition. This pattern is also similar to that seen with deltamethrin-based nets (PermaNet 2.0 vs. PermaNet 3.0) [29,31]. Overall, the significantly greater loss of efficacy due to P450s observed with both type I (Olyset and PermaNet 2.0) and PBO-based nets (Olyset Plus and PermaNet 3.0) further supports the deployment of PBO-based nets to control P450-based metabolically resistant mosquito populations. The deployment of PBO-based nets should nevertheless also be monitored to regularly assess their efficacy since they too could be impacted by resistance, as shown by the fact that *CYP6P9a/b*-resistant mosquitoes could blood-feed significant better, even with Olyset Plus. This increased ability of resistant mosquitoes to blood-fed, even with PBO-based net, suggests that PBO-based nets are not immune to the impact of resistance, even if they remain significantly more effective than pyrethoid-only nets. The increased blood-feeding of P450-resistant mosquitoes when exposed to PBO-based nets may also lead to a higher malaria transmission, although this is mitigated by the high mortality observed for Olyset Plus in a experimental hut trial. Overall, evaluating the impact of P450-based resistance on the efficacy of Olyset vs. Olyset Plus further supports the results obtained in a cluster-randomized controlled trial with both nets in Tanzania showing greater effectiveness of Olyset Plus [46].

### 3.3. Resistance Escalation with Multiple Resistance Alleles Present a Greater Risk of Control Failure

This study confirms the findings by [28], suggesting that the *6.5 kb SV* acts as an enhancer for nearby duplicated P450 genes *CYP6P9a* and *CYP6P9b*, leading to their increased overexpression, thus creating greater resistance. This *6.5 kb SV* is strongly associated with an aggravation of pyrethroid resistance, which reduces the efficacy of pyrethroid-only nets. The fixation of this *6.5 kb SV*, besides the resistant alleles of *CYP6P9a* and *CYP6P9b*, could explain the resistance escalation currently observed against the *An. funestus* population from the southern part of Africa reducing the efficacy of bed nets [7,41]. 

This study revealed that multiple and complex resistance combining elevated expression (*CYP6P9a/b*) [45], as well as the selection of cis-regulatory motifs for transcription binding sites (CnCC/MAF) [31], coupled with structural variations such as the *6.5 kb*, which is known to be enriched with several regulatory elements [28], can lead to a greater reduction in bed net efficacy. The fact that triple homozygote-resistant mosquitoes could survive better against exposure to pyrethroid-only nets compared to all genotypes reveals the greater risk that an unabated increase in resistance can likely lead to the failure of insecticide-based interventions, as predicted by the WHO global plan for insecticide resistance management. This calls for novel nets which do not rely on pyrethroids in the future. However, the impact of these resistance alleles on the efficacy of LLINs in natural populations remains to be established, as we only performed a test with a hybrid strain from two laboratory strains. This work must be urgently carried out, particularly as the molecular tools are increasingly available.

## 4. Materials and Methods

### 4.1. Study Site

The experimental station is located at Mibellon (6°4′60″ N, 11°30′0″ E), a village in Cameroon located in the Adamawa region, in the Mayo Banyo division and Bankim sub-division. The Adamawa region is in a mountainous area, forming a transition between Cameroon’s forested south and savanna north. At the experimental station, 12 huts are built, following WHO standards [47], which are available for a wide range of experimental hut trials, as previously described [43]. The trial was carried out in August 2017.

### 4.2. Laboratory Strain: FUMOZ/FANG Crossing

Previous studies revealed that the duplicated P450 *CYP6P9a/b* and intergenic *6.5 kb sv* is mainly found in mosquitoes from southern Africa and absent from mosquitoes collected elsewhere in Africa [31,45]. Moreover, these resistance markers have already been selected (close to fixation) in the southern African *An. funestus* populations, preventing free-flying mosquitoes from being used in this region in order to assess their impact on LLIN efficacy, as all mosquitoes are nearly homozygote-resistant now [7]. Therefore, to evaluate the impact of these cytochrome P450, we opted to use a hybrid strain generated from two *An. funestus* laboratory colonies: the FANG colony, a completely insecticide-susceptible colony originating from Angola, and the FUMOZ colony derived from southern Mozambique, which is highly resistant to pyrethroids and carbamates [48]. During rearing, the pupae of each strain were kept individually in Falcon tubes (15 mL), locked with a piece of cotton for individual emergence. After the emergence of pupae in the Falcon tube, the males and the females were separated. A reciprocal crossing was performed using 50 males and 50 females from the other strain. After the initial F_1_ generation obtained from the reciprocal crosses of the 50 males and 50 females of both strains, the hybrid strain was reared to F_5_ and F_6_ generation, presenting good segregation in all the three genotypes. The hybrid strain was then used for the release recapture experiment in the huts.

### 4.3. Susceptibility Profile of the Hybrid FUMOZ/FANG Strain to Pyrethroids

Before assessing the impact of *CYP6P9a/b* on the effectiveness of LLINs using the experimental huts, the resistance status of the hybrid FUMOZ-FANG was evaluated in laboratory conditions via WHO tube tests and cone tests, and the role of *CYP6P9a/b* in terms of resistance was assessed.

**Bioassays:** WHO bioassays were conducted with 0.05% deltamethrin and 0.1% propoxur. To generate highly resistant and highly susceptible mosquitoes, WHO bioassays were conducted with 0.75% permethrin and 0.05% deltamethrin for 30 min and 90 min. For each insecticide, four replicates of 25 mosquitoes from the hybrid strains were used. Alive mosquitoes after 90 min of exposure and dead mosquitoes after 30 min of exposure were then genotyped to establish the association between the *CYP6P9a/b* and *6.5 kb SV* resistance alleles and the ability of mosquitoes to survive to these insecticides.

**Cone assays:** Cone test bioassays were conducted with a fragment of Olyset and Olyset Plus using the resistant hybrid strains FUMOZ-FANG. Five batches of 10 unfed females, aged 2–5 days old, were exposed to each bed net for three minutes. They were then transferred into the holding paper cup containers. The knockdown was checked after 60 min and the mortality after 24 h. Alive and dead mosquitoes obtained after exposure were then genotyped. The association between the *CYP6P9a/b*-resistant allele and the ability of mosquitoes to survive to these insecticides was also established.

### 4.4. Experimental Hut Design

The huts were built following the prototype recommended by WHO for the West African region, as previously described [43]. The hut was constructed on a concrete base of cement surrounded by a drain channel to trap the aunt. The walls were made from concrete bricks and were plastered inside and outside with plaster made from a mixture of cement and sand. The roof was made from corrugated iron and the ceiling was made from plywood. In the context of this trial, since we released mosquitoes in the huts, all the window openings were closed to avoid the mosquitoes from escaping.

#### 4.4.1. Hut Treatment/Arm Comparison

During the experimental hut trial, 2 insecticide-treated nets, kindly provided by Sumitomo, and one untreated net were compared (Table 1):

#### 4.4.2. Quality Control of Bed Nets Used in the Study

Cone bioassays were performed at the beginning of the study using the Kisumu-susceptible laboratory strain to confirm the quality of the three bed nets involved in the study. Five batches of 10 unfed females, aged 2–5 days old, were exposed to each bed net using WHO cone assays for three minutes. They were then transferred into the holding paper cup containers. The knockdown was checked after 60 min and the mortality after 24 h. 

#### 4.4.3. The Experimental Hut Trial 

The experimental hut trial was carried out using the release–recapture approach by releasing the hybrid lab strain FUMOZ-FANG in the treated hut every evening. 

**Latin square design rotation and volunteer sleepers**: Three volunteering men were selected to sleep in the room from 20:00 GMT to 5:00 GMT in the morning. One volunteer slept under an Olyset net, one under an Olyset Plus net, and the third volunteer under the untreated net. Since a fixed number of mosquitoes were released every night in the huts selected for the study, a rotation of sleepers through a Latin square design was not necessary in this study. This design is usually applied to correct any specific attractiveness that some huts may have due to their position [47].

**Releasing and recapture**: During the test night, volunteers were installed in the different huts selected for the study. Then, 50 to 100 female mosquitoes from the lab strain FUMOZ-FANG were equally released in the huts for a total of 6 days. Mosquitoes were collected every morning, using hemolysis tubes from: (i) inside the nets; (ii) in the room: floor, walls, and roof; and (iii) in the veranda exit trap. Mosquitoes collected from each compartment were kept separately in a bag to avoid any mix-up between samples from different compartments. Samples were transferred to the laboratory for identification. After identification, samples were then classified as dead, alive, blood-fed, or unfed. The ‘alive’ samples were kept in the paper cup and provided with sugar solution for 24 h and their mortality was monitored. Dead mosquitoes were then kept adequately in labelled Eppendorf tubes with Silicagel, and live mosquitoes were stored in RNA later.

### 4.5. Bed Nets Performance Assessment 

The performance of the bed nets was expressed relative to control (untreated nets). This was performed with four parameters in mind.

**(i)** **Exophily**. The proportion of mosquitoes found exited in the veranda trap Exophily (%) = 100 × (Ev/Et), where Ev is the total number of mosquitoes found in veranda and Et is the total number of mosquitoes in the hut.**(ii)** **Blood-feeding rate (BFR).** This rate was calculated as follows: blood-feeding rate = (N mosquitoes fed) × 100/total N mosquitoes, where N mosquitoes fed was the number of mosquitoes fed and a total N mosquito was the total number of mosquitoes collected.**(iii)** **Blood-feeding inhibition (BFI)**. The reduction in blood-feeding in comparison with the control hut. Blood-feeding inhibition is an indicator of personal protection (PP). More precisely, the personal protection effect of each bed net is the reduction in the blood-feeding percentage induced by the net when compared to the control. The protective effect of each bed net can be calculated as follows: *personal protection* (%) = 100 × (*Bu* – *Bt*)/*Bu*, where *Bu* is the total number of blood-fed mosquitoes in the huts with untreated nets and *Bt* is the total number of blood-fed mosquitoes in the huts with treated nets [47].**(iv)** **Immediate and delay mortality**. The proportion of mosquitoes entering the hut that are found dead in the morning (immediate mortality) or after being caught alive and held for 24 h with access to sugar solution (delay mortality) [47]. In this study, we presented the overall mortality calculated as follows: mortality (%) = 100 × (Mt/MT), where Mt is the total number of mosquitoes found dead in the hut and MT is the total number of mosquitoes collected in the hut [28,47].**(v)** As mosquitoes were rather released in the huts, the deterrence, i.e., the reduction in the entry rate of mosquitoes in the treated huts relative to control, could not be determined here.

### 4.6. Impact of the Duplicated CYP6P9a and CYP6P9b P450 Genes on the Performance of Bed Nets

The samples collected during the evaluation of Olyset and Olyset Plus in experimental huts were grouped in several categories: dead, alive, blood fed, and unfed; room and veranda. The *CYP6P9a*/b and *6.5 kb SV* were genotyped in each group using the respective PCR assays as described [28,29,31]. This allowed the relative survival and feeding success of resistant and susceptible insects in the presence of both bed nets to be directly measured.

#### 4.6.1. Genotyping of the CYP6P9a-R Marker Using PCR-RFLP

DNA was extracted from various groups of mosquitoes (dead, alive, blood fed, and unfed; room and veranda) using the Livak protocol [49]. The PCR was carried out using 10 mM of each primer and 1 µL of gDNA as the template in 15 µL reaction containing 10× Kapa Taq buffer A, 25 mM of dNTPs, 25 mM of MgCl2, and 1 U of Kapa Taq (Kapa Biosystems, Boston, MA, USA). Amplification was carried out using thermocycler parameters: 95 °C for 5 min, 35 cycles of 94 °C for 30 s, 58 °C for 30 s, 72 °C for 45 s, and a final extension at 72 °C for 10 min. The following primers were used: RFLP6P9aF forward primer 5′-TCCCGAAATACAGCCTTTCAG-3 and RFLP6P9aR reverse primer 5′-ATTGGTGCCATCGCTAGAAG-3′. Then, 3 μL of the PCR product migrated on the 1.5% agarose gel. The expected PCR product was 450 bp. The digestion with Taq1α followed 0.2 μL of Taq1α, 5 μL of PCR product, 1 μL of 10× NEBuffer, and 3.8 μL of distilled water. This mix was incubated at 65 °C for 2 h. Afterwards, 3 μL of the digestion product migrated on the 2% gel. Amplicons from resistant mosquitoes were cut into two bands with sizes of 350 bp and 100 bp, and the susceptible mosquitoes remained undigested with the band at 450 bp.

#### 4.6.2. Genotyping of the CYP6P9b-R Maker Using PCR-RFLP

The PCR was carried out, as described for CYP6P9a. The following primers were used: 6p9brflp_0.5F 5′-CCCCCACAGGTGGTAACTATCTGAA-3′ and 6p9brflp_0.5R 5′-TTATCCGTAACTCAATAGCGATG-3′. Then, 3 μL of the PCR product migrated on the 1.5% agarose gel. The expected PCR product was 550 bp. The digestion with the *Tsp*451 restriction enzyme followed 0.2 μL of Tsp451, 5 μL of PCR product, 1 μL of 10× NEBuffer, and 3.8 μL of distilled water. This mix was incubated at 65 °C for 2 h. Afterwards, 3 μL of the digestion product was migrated on the 2% gel. Amplicons from susceptible mosquitoes were cut into two bands with sizes of 400 bp and 150 bp, and the resistant mosquitoes remained undigested with the band at 550 bp.

#### 4.6.3. PCR Assay to Detect the 6.5 kb SV

To easily identify the samples containing the 6.5 kb insertion, the recently designed PCR assay [28] was used to discriminate between mosquitoes with the 8.2 kb (resistant) and 1.7 kb (susceptible) *CYP6P9a* and *CYP6P9b* intergenic regions. Briefly, three primers were used: two (FG_5F: CTTCACGTCAAAGTCCGTAT and FG_3R: TTTCGGAAAACATCCTCAA) at regions flanking the insertion point and a third primer (FZ_INS5R: ATATGCCACGAAGGAAGCAG) in the *6.5 kb* insertion. One unit of KAPA Taq polymerase (Kapa Biosystems) in 1× buffer A, i.e., 25 mm of MgCl_2_, 25 mm of dNTPs, and 10 mm of each primer, was used to constitute a 15 μL PCR mix using the following conditions: an initial denaturation step of 3 min at 95 °C, followed by 35 cycles of 30 s at 94 °C, 30 s at 58 °C, and 60 s at 72 °C, with a final extension for 10 min at 72 °C. The amplicon was revealed on a 1.5% agarose gel stained with Midori Green Advance DNA Stain (Nippon genetics Europe GmbH) and revealed on a UV transilluminator.

### 4.7. Statistical Analysis

**Experimental hu****t trial**: To calculate the proportion of each entomological outcomes and the level of significance between the treatments and between the control for each entomological outcomes, the XLSTAT software was used, as before. The confidence limits of these parameters were calculated and the proportions were compared using a chi-square test with a significance limit of 0.05 [31,43,50,51].

**Test of association between the P450 genes (*CYP6P9a/b*) and the entomological outcomes:** To investigate the association between the P450 alleles and mosquito’s ability to survive, blood-feed, or exit the room with bed nets, VassarStats [52] was used to estimate the odds ratio based on a fisher exact probability test with a 2 × 2 contingency table, as previously described [31,43].

**Ethical clearance**: The national ethics committee for health research of Cameroon approved the protocol of the study (ID: 2016/03/725/CE/CNERSH/SP). Written, informed, and signed consent was obtained from sleepers before starting the trials. The consent form provided all the information and the evaluation process regarding the study. Information was translated in a local language when needed. All the volunteers involved in the study were followed-up and treated when they presented malaria symptoms. All methods were performed in accordance with the relevant guidelines and regulations.

## 5. Conclusions

This study reveals that insecticide resistance driven by *CYP6P9a*/*b* and *6.5 kb SV* can reduce the efficiency of Permethrin-based LLINs, such as Olyset, while PBO-based nets remain effective. However, the greater loss of efficacy observed in mosquitoes with multiple and complex resistance supports the need to introduce new products for vector control which is less reliant on pyrethroids. Meantime, PBO-based nets should be preferably deployed in areas of P450-based resistance.

## Figures and Tables

**Figure 1 pathogens-11-00638-f001:**
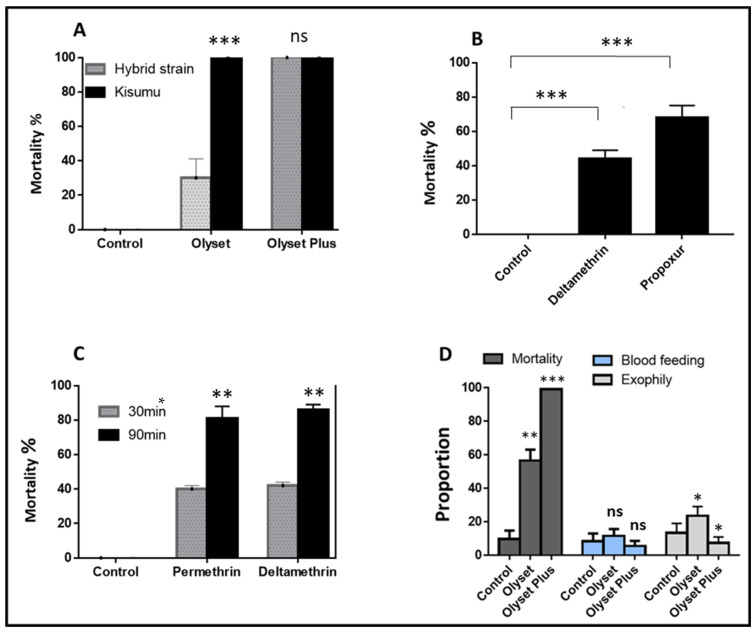
Susceptibility profile of the hybrid FUMOZ-FANG strain to pyrethroids. (**A**) Net quality assessment with recorded mortalities after cone assays with Kisumu, the *An. gambiae*-susceptible lab strain, and the *An. funestus* FUMOZ-FANG strain. (**B**) Susceptibility profile of the hybrid FUMOZ-FANG strain to type II (deltamethrin) pyrethroids and propoxur with recorded mortalities following 60 min exposure. Data are shown as mean ± SEM. (**C**) Susceptibility profile of the hybrid FUMOZ-FANG strain to deltamethrin (type II pyrethroids) and permethrin (type I pyrethroids) with recorded mortalities following 30 and 60 min exposure. Data are shown as mean ± SEM. (**D**) Proportion of mortality, blood-feeding, and exophily rate for Olyset and Olyset Plus against *An. funestus* (crossing the FUMOZ-FANG strain (F5)). Ns = *p* > 0.05; * = *p* ≤ 0.05; ** = *p* ≤ 0.01; *** = *p* ≤ 0.001.

**Figure 2 pathogens-11-00638-f002:**
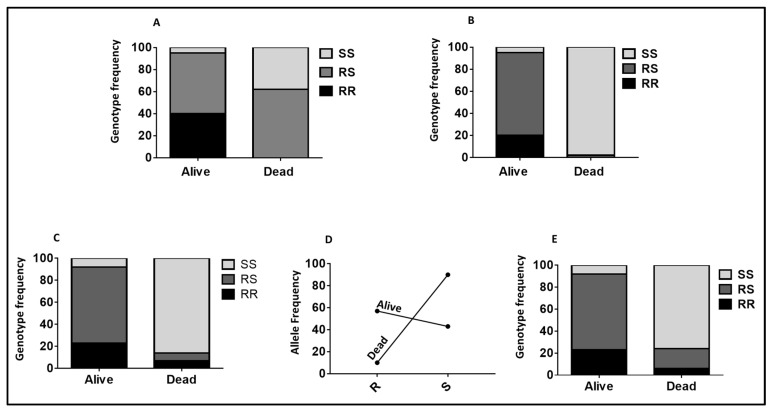
Association between the *CYP6P9a/b* and the ability to survive exposure to Olyset nets after WHO tubes cone assays. (**A**) Tube assay *CYP6P9a* and mortality. Role of *CYP6P9a* in pyrethroid resistance with samples from the WHO tube. Distribution of the *CYP6P9a* genotypes according to resistance phenotypes. (**B**) Tube assay *CYP6P9b* and mortality. Role of *CYP6P9b* in pyrethroid resistance with samples from the WHO tube. Distribution of the *CYP6P9b* genotypes according to resistance phenotypes. (**C**) Cone assay *CYP6P9a* and mortality. Genotype distribution of *CYP6P9a* between alive and dead mosquitoes after exposure to Olyset with the cone test, showing the significant ability of mosquitoes carrying the RR genotype in order to survive compared to SS (RR vs. SS: *p* < 0.001). (**D**) Cone assay *CYP6P9a* and mortality allele. Allelic frequency of *CYP6P9a* between alive and dead mosquitoes after exposure to Olyset with the cone test, showing the significant ability of the *CYP6P9a*_R_-resistant mosquitoes to survive compared to their susceptible counterparts *CYP6P9a*_S. (**E**) Cone assay *CYP6P9b* and mortality. Genotype distribution of *CYP6P9b* between alive and dead mosquitoes after exposure to Olyset using the cone test, showing strong association (RR vs. SS: *p* < 0.001).

**Figure 3 pathogens-11-00638-f003:**
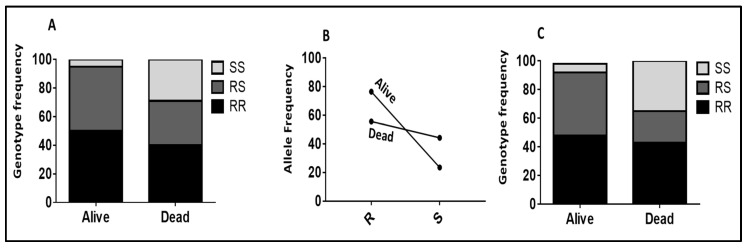
Association between the *CYP6P9a/b* and the ability to survive exposure to Olyset Plus nets after cone assays. (**A**) Genotype distribution of *CYP6P9a* between alive and dead mosquitoes after exposure to Olyset Plus with the cone test, showing the significant ability of the mosquitoes to carry the RR genotype to survive compared to SS (*p* < 0.001). (**B**) Allelic frequency of *CYP6P9a* between alive and dead mosquitoes after exposure to Olyset Plus with the cone test, showing the significant ability of the *CYP6P9a*_R_-resistant mosquitoes to survive compared to their susceptible counterparts *CYP6P9a*_S. (**C**) Genotype distribution of *CYP6P9b* between alive and dead mosquitoes after exposure to Olyset Plus using the cone test, showing the significant ability of the mosquitoes carrying the RR genotype to survive compared to SS (*p* < 0.001).

**Figure 4 pathogens-11-00638-f004:**
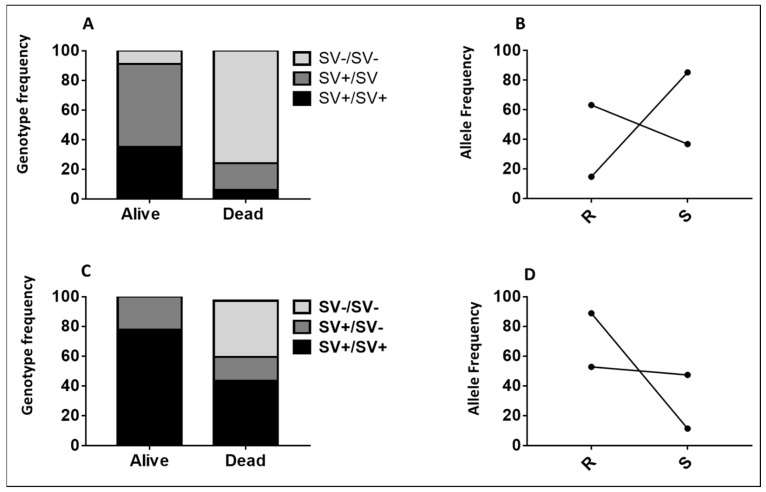
Association between 6.5 kb SV and the ability to survive against exposure to Olyset and Olyset Plus nets using cone assays. (**A**) Distribution of *6.5 kb SV* genotypes between dead and alive mosquitoes after exposure to Olyset, showing that *6.5 kb SV*_R allows mosquitoes to survive significantly better against exposure to this net. (**B**) Allelic frequency of *6.5 kb SV* between alive and dead mosquitoes after exposure to Olyset with the cone test, showing the significant ability of the *6.5 kb SV* _R_-resistant mosquitoes to survive compared to their susceptible counterparts *6.5 kb* SV _S. (**C**) Distribution of *6.5 kb SV* genotypes between dead and alive mosquitoes after exposure to the Olyset Plus net. (**D**) Allelic frequency of *6.5 kb SV* between alive and dead mosquitoes after exposure to Olyset Plus with the cone test, showing the significant ability of the *6.5 kb SV* _R_-resistant mosquitoes to survive compared to their susceptible counterparts *6.5 kb SV* _S.

**Figure 5 pathogens-11-00638-f005:**
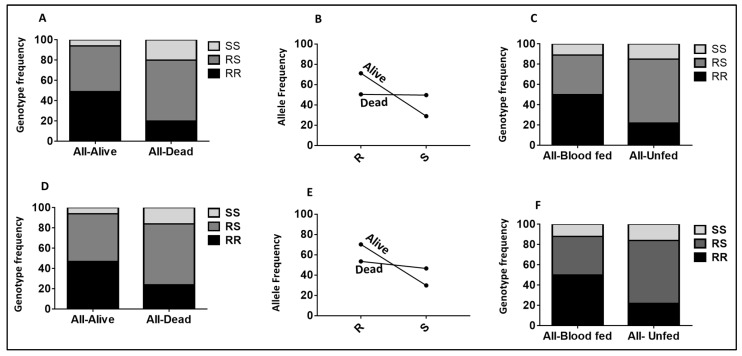
Impact of both *CYP6P9a* and *CYP6P9b* on the efficacy of bed nets in the experimental hut trial. (**A**) Genotype distribution of *CYP6P9a* between alive and dead mosquitoes after exposure to Olyset, showing strong association (RR vs. SS: *p* < 0.001). (**B**) Allelic frequency of *CYP6P9a* between alive and dead mosquitoes after exposure to Olyset with experimental hut trial, showing the significant ability of the *CYP6P9a*_R_-resistant mosquitoes to survive compared to their susceptible counterparts *CYP6P9a*_S. (**C**) Association between *CYP6P9a* and ability to blood-feed when exposed to Olyset. The *CYP6P9a*-R allele increases the strength of resistant mosquitoes by taking a blood meal in contrast to the susceptible ones when exposed to Olyset net. (**D**) Association between *CYP6P9b* and ability to survive exposure to Olyset in the experimental hut trial. (**E**) Allelic frequency of *CYP6P9b* between alive and dead mosquitoes after exposure to Olyset with the experimental hut trial, showing the significant ability of the *CYP6P9b_*R_-resistant mosquitoes to survive compared to their susceptible counterparts *CYP6P9b*_S. (**F**) Association between duplicated *CYP6P9b* gene and ability to blood-feed when exposed to Olyset in the experimental hut trial. The *CYP6P9b*-R allele increases the strength of resistant mosquitoes by taking a blood meal in contrast to the susceptible ones when exposed to Olyset.

**Figure 6 pathogens-11-00638-f006:**
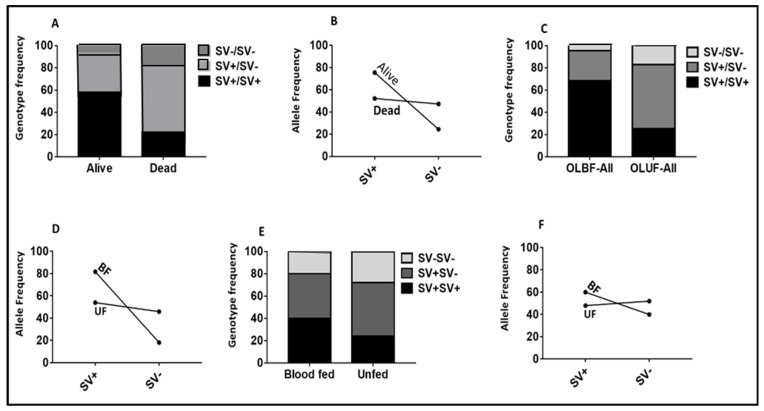
Impact of the *6.5 kb SV*-based metabolic resistance on bed nets’ efficacy in experimental hut trials. (**A**) Distribution of *6.5 kb SV* genotypes between dead and alive mosquitoes after exposure to Olyset net, showing that *6.5 kb SV* _R allows mosquitoes to survive significant better after exposure to this insecticide-treated net. (**B**) Allelic frequency of *6.5 kb SV* between alive and dead mosquitoes after exposure to Olyset with experimental hut trial, showing the significant ability of the *6.5 kb SV* _R_-resistant mosquitoes to survive compared to their susceptible counterparts *6.5 kb SV* _S. (**C**) Distribution of *6.5 kb SV* genotypes between the blood fed (BF) and unfed (UF) mosquitoes after exposure to Olyset (OL), showing that *6.5 kb SV*_R significantly allows mosquitoes to blood-feed in the presence of this insecticide-treated net. (**D**) Allelic frequency of *6.5 kb SV* between alive and dead mosquitoes after exposure to Olyset with the experimental hut trial, showing the ability of the *6.5 kb SV* _R_-resistant mosquitoes to blood-feed significantly better compared to their susceptible counterparts *6.5 kb SV* _S. (**E**) Distribution of *6.5 kb SV* genotypes between blood-fed (BF) and unfed (UF) mosquitoes after exposure to Olyset Plus, showing that *6.5 kb SV*_R significantly allows mosquitoes to blood-feed in the presence of this insecticide-treated net. (**F**) Allelic frequency of *6.5 kb SV* between alive and dead mosquitoes after exposure to Olyset Plus with the experimental hut trial, showing the significant ability of the *6.5 kb SV* _R_-resistant mosquitoes to blood-feed compared to their susceptible counterparts *6.5 kb SV* _S.

**Figure 7 pathogens-11-00638-f007:**
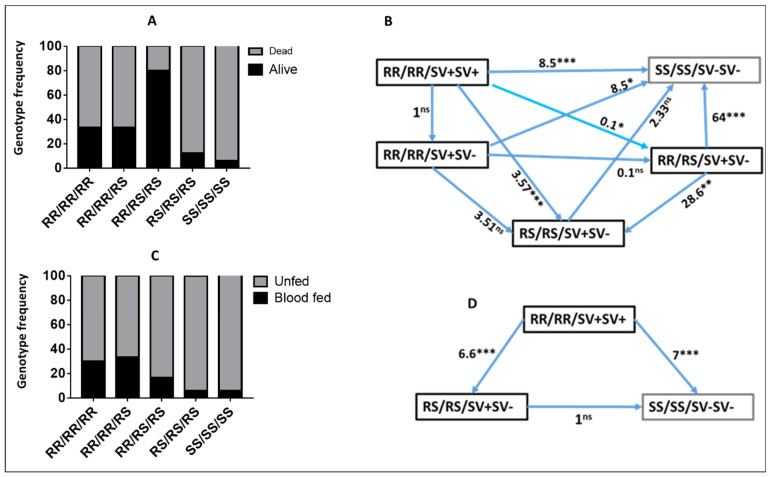
Combined impact of the triple resistance alleles (*CYP6P9a/CYP6P9b/6.5 kb SV*) on the efficacy of insecticide-treated nets. (**A**) Distribution of combined genotypes for *CYP6P9a, CYP6P9b* and the *6.5 kb SV* showing that the three markers combine to increase the chances of being alive in the presence of Olyset Net. (**B**) Ability to survive exposure to Olyset net for the various combined genotypes for the *6.5 kb SV, CYP6P9a* and *CYP6P9b*. (**C**) Distribution of combined genotypes for *CYP6P9a, CYP6P9b* and the *6.5 kb SV* showing that the three markers combine to increase the chances to bloodfed in the presence of Olyset net. (**D**) The triple RR/RR/SV+SV+ homozygous mosquitoes for the *6.5 kb SV, CYP6P9a* and *CYP6P9b* exhibit a greater blood-feeding ability than other genotypes. Ns: non-significant; * (0.05); ** (0.01); *** (0.001).

**Table 1 pathogens-11-00638-t001:** Description of the long-lasting insecticidal nets used.

Treatment Arm	Description	Manufacturer
Untreated	100% polyester with no insecticide	Local market
Olyset	8.6 × 10^−4^ kg/m^2^ (2%) of permethrin incorporated into polyethylene	Sumitomo Chemical
Olyset Plus	8.6 × 10^−4^ kg/m^2^ (2%) of permethrin and 4.3 × 10^−4^ kg/m^2^ (1%) of PBO incorporated into polyethylene	Sumitomo Chemical

## Data Availability

NA.

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
