# Peer review of "Experimental Hut Trials Reveal That CYP6P9a/b P450 Alleles Are Reducing the Efficacy of Pyrethroid-Only Olyset Net against the Malaria Vector Anopheles funestus but PBO-Based Olyset Plus Net Remains Effective"

_pathogens, 2022, doi:10.3390/pathogens11060638_

Round 1

Reviewer 1 Report

Insecticide resistance monitoring is essential in regions where malaria transmission occurs. Understanding the mechanisms conferring resistance is necessary for program implementation and in managing the development of resistance. In this study, a hybrid strain of A. funestus was reared such that differential insecticide susceptibility and genotype was observed for study purposes. The authors set up an experimental hut trial in order to test the contribution of different alleles and genetic markers on resistance status. The study was well executed, thorough and the findings relevant. In some instances, too much emphasis was placed on relevance a finding – in particular, the propensity of homozygous resistant mosquitoes to blood feed over other genotypes. Please see specific comments below.

Spelling, grammar and typos must be corrected throughout.

Materials and methods:

Line 562: Add the following to this paragraph -Hut set-up: One volunteer slept under an Olyset net, one under an Olyset Plus net and the third volunteer under the untreated net.

Release and recapture:

Line 567: I assume that only females were released into each hut? Please clarify.

Statistical analysis:

Line 653: Experimental Hut Trial – please include a description of the statistical methods rather than cite other papers.

  1. Results

Figure 1:

Please include the statistical outcomes on the figures.

Table 2:

I would suggest removing this table (or putting it in Supplementary material section) – it is somewhat confusing and I don’t know if it adds much value over and above the figures.

2.3.2 and 2.4.2 CYP6P9a and CYP69b impact on blood feeding:

Surely given the fact that most susceptible (SS) mosquitoes will die, it is logical that blood-feeders are RR? I would say that this result is largely a numbers game. This must be addressed in the discussion as I am not convinced that any conclusions can really be drawn here.

2.7. & Figure 7A: As I understand this figure, the genotype RR/RS/RS is the optimal genotype for surviving bednet exposure? This is not in agreement with section 2.7 as it is written. Please clarify or correct as this is in disagreement with all data that has come before.

Author Response

Response to Reviewer 1 Comments

Point 1: Spelling, grammar and typos must be corrected throughout.

Response 1: Thanks very much for these remarks, the spelling, the grammar and the typos have been corrected throughout the manuscript

Materials and methods:

Point 2: Line 562: Add the following to this paragraph -Hut set-up: One volunteer slept under an Olyset net, one under an Olyset Plus net and the third volunteer under the untreated net.

 Response 2: These clarifications have been added as suggested

Point 3: Release and recapture: Line 567: I assume that only females were released into each hut? Please clarify.

 Response 3: The clarification has been done at the line 567. Please see the version with track change

Point 4: Statistical analysis: Line 653: Experimental Hut Trial – please include a description of the statistical methods rather than cite other papers.

 Response 4: More details regarding the statistical analysis have been added. Please see the track change version from line 653.

Point 5: Results, Figure 1: Please include the statistical outcomes on the figures.

Response 5: The statistical outcomes have been added on Figure 1. Please, see the track change version

Point 6: Table 2: I would suggest removing this table (or putting it in Supplementary material section) – it is somewhat confusing and I don’t know if it adds much value over and above the figures.

 Response 6: Table two has been sent to supplementary files as requested and the manuscript has been changed accordingly.

Point 6: 2.3.2 and 2.4.2 CYP6P9a and CYP69b impact on blood feeding:

Surely given the fact that most susceptible (SS) mosquitoes will die, it is logical that blood-feeders are RR? I would say that this result is largely a numbers game. This must be addressed in the discussion as I am not convinced that any conclusions can really be drawn here.

 Response 6: Thanks very much for this comment. It always seems like it is logical that blood feeders are RR and that most of susceptible (SS) will die. But this is not true for all the markers. A different marker may show that the SS are not dying or the blood-feeders are not RR and it is not rare to observe this. So the fact that this correlation is clear for this marker in this population establishes the fact that these markers are impacting the effectiveness of vector control tools for this population.

Point 7: 2.7. & Figure 7A: As I understand this figure, the genotype RR/RS/RS is the optimal genotype for surviving bednet exposure? This is not in agreement with section 2.7 as it is written. Please clarify or correct as this is in disagreement with all data that has come before.

Response7: I agree with you. This portion has been corrected. Please see the track change version.

Reviewer 2 Report

comments attached

Author Response

Response to Reviewer 2 Comments

Point 1: Line 25-26 “… revealing a greater loss of efficacy for the latter”[Olyset Plus] – You mean the former (Olyset)? This seems misleading given that Olyset Plus had 99.1% mortality rate than Olyset (56.7%). Without the elaboration of what is the loss of efficacy means, this can be conflicting statement.

Response 1: I agree with you that this statement is confusing. This has been clarified, see the track change version.

Point 2: Line 38, 82: use EHT if you are going to use the acronyms introduced in line 23.

Response 2: This has been modified in the text

Point 3: Line 62-63: the choice of word ‘target’ is vague and can be source of confusion. I suggest to add a word like target organism, target gene, target protein, etc. for clarification.

Response3: This has been clarify in the manuscript: line 62

Point 4: Line 65-67: kdr is italicized in some places and not in other places. Please use consistent typesetting.

Response4: Thanks for the remark, this has been corrected in the manuscript

Point 5: Line 71-74: citation missing for the referred recent studies.

Response 5: References have been added: line 74

Point 6: Line 75-79: Break into two sentences.

Response 6: This has been done

Point 7: Introduction – It would be helpful to know how commonly Olyset and Olyset Plus (and perhaps PermaNet since it is mentioned in the paper) is used in which countries, the proportions of its market share if known. This information will be able to tie better with the impact of this study. Also information about the relative abundance of the genotypes in natural population also seems to be missing.

Response 6: Difficult to know the proportions of the market share for these nets in countries. However in Cameroon, during the last mass campaign distribution of bet nets, Olyset and Olyset Plus were about 40% of the nets involved in the campaign. This scientific article is really focus on the P450 (CYP6P9a/b and SV) impacting the effectiveness of Olyset and Olyset Plus

Point 7: Figure 1D – I understand the value of y axis is proportion but it seems better to separate blood feeding and exophily in separate panel with their own scale as they measure different things. It is misleading to present them in its current form – as it can indicate that blood feeding and exophily are not much impacted by the net type – which is contrary to the results.

Response 7: For Olyset and Olyset plus the blood feeding and the exophily rate did not really differ compare to control. However when looking at the mortality we notice a highly significant difference between control vs Olyset and Olyset Plus. I am not sure if putting them in different graph will give a different trend.

Point 8: Figure 1C – I don’t see 30min vs 90min is that important or relevant – not mentioned in the abstract or intro. It is also confusing why 1B used 60 min exposure with Deltamethrin and propoxur but the 1C used permethrin and deltamethrin.

Response 8: The aim of testing 30min vs 90min is to generate highly susceptible and highly resistant mosquitoes. It is mainly for a purpose of generating samples with different phenotype. The WHO protocol normally requires 60 minutes exposition. So before changing the exposition time to generate samples with different phenotype, it is always good to present first results with 60 minutes exposure. This is reason why we decided to present 60 minutes results against pyrethroid and another insecticides class.

Point 9: Figure 1B – I think it should include profile of the original parental strains to understand the hybrid strain susceptibility profile.

Response 9: All the test in this study were done with F5-F6 crossing from to lab strain. Line 512 to 521 explain well the resistance status of the parental strain.

Point 10: Figure 2 – Put some subheading to make the figure self-explanatory. E.g. A-tube assay - CYP6P9a, B-tube assay - CYP6P9b, C: Olyset – CYP6P9a, E- Olyset-CYP6P9b.. Also swap place for D and E.

Response 10: subheadings have been put as requested. Difficult to swap E and D since it needs to appear in the same oder like I the manuscript.

Point 11: Figure 2D - Linear line is also misleading when comparing dead vs alive. For readers skimming the content, the notion of allele frequency of dead vs alive also doesn’t make much sense. If you want to show high proportion of resistant alleles in alive group, why not simple bar chart with resistant and susceptible allele frequency (sum up to 1) for Alive and Dead group? That’s much easier to understand figure.

Response 11: I understand your point. This graph also simply and clearly shows how the frequency differs between dead and alive. Please this format is very common to us:

Mugenzi et al 2020, Nature communications

Mugenzi et al 2020, Molecular ecology

Point 12: Section 2.2.3 – rationale for using multiple time points and what the results mean is not well explained.

Response 12: The aim of testing 30min vs 90min is to generate highly susceptible and highly resistant mosquitoes. It is mainly for a purpose of generating samples with different phenotype.

Point 13: Figure 3B – see Figure 2D concern.

Figure 4B and D – see Figure 2D concern. Which one is alive or dead is not clearly noted.

Figure 5 – same as Figure 2D concern.

Figure 6 – same as Figure 2D concern.

Response 13: Please, see response 11

Point 14: Line 140 – “CYP6P9a is additive” – Olyset cone test results seems to consistent with this statement but I don’t think sufficient evidence is presented to indicate that this is additive or multiplicative or any other relationship in general. About equal ratio of individuals with RS genotype were presented in both alive and dead when exposed to WHO tube assay. It seems like the CYP6P9a resistant allele is more like recessive – you have to have two copies to be able to have chance to be alive. Perhaps clarify the conditions where this odd ratio is calculated and do not overgeneralize.

Response 14: I agree with you that using additive can be confusing. The word additive was replaced

Point 15: Section 2.2.3 – Can you say comparable notes (additive or other mode?) for CYP6P9b? What about dominance (dominant vs recessive?)

Response15: What we would like to communicate here is just that if you are RR for this marker you are more able to survive exposure to insecticide.

Point 16: Line 234-242: I think the sentences should be rearranged to flow better. Currently the sentences are arranged so that 1) RR showed increased survivalship. 2) RS did not survive any better SS. 3) ‘Moreover,’ R provide higher resistance than S. This should be rearranged so that 1) higher survival of RR, 2) R better than S for survival, 3) “However,” RS did not survive any better than SS. (and perhaps this would be natural place to add that R seems to be recessive).

Response 16: Thank for these suggestions, this portion was rearranged. Please, see the track change version

Point 17: Line 264-269: Perhaps mention that CYP6Pb is dominant allele?

Response 17: I would say co-dominant

Point 18: Figure 6A and C – some outlines of bar graph are missing resulting in inconsistent format of figures.

Response 18: The figure has been reformatted. Please see the version with track change

Point 19: Results section – I wonder if differentiating R to Ra (CYP6P9a) and Rb (CYP6P9b) would be helpful to readers to track which gene is being discussed when reading this paper. This especially applies to section 2.6.

Response 19: Thank you for this suggestion. I think if you read carefully it will be very easy to notice that the first is for a and the second for b

Point 20: Figure 7A and C – Outline for the last bar is missing.

Response20: This has been added

Point 21: Line 444-446 – reference missing to support the sentence.

Response 21: reference has been added

Point 22: Discussion – genomic location of 2 genes and 1 genomic segment and linkage or linkage disequilibrium are relevant topic to discuss as all shows similar trend but missing. Also it is now clear how widespread the mutations studied in natural populations and the typical nets used in Africa. If not known, then it should be also listed as future study to assess the extent of insecticide resistance problem and perhaps inform WHO, US PMI, and other agencies that are involved in purchasing those nets.

Response 22: Thanks for this comment

Point 23: Line 526-527 – Figure of WHO tube test, and olyset cone test would be helpful to readers who are not familiar with this type of insecticide resistance test. It is not clear why mixed test were used. Are results comparable or interchangeable? Why not use a single method? Explain the rationale or circumstances on why two methods had to be used.

Response23: Usually the tube test is to assess the susceptibility profile of a population and the cone test is to assess the impact of that profile on the performance of the nets.
